# Periostin from Tumor Stromal Cells Might Be Associated with Malignant Progression of Colorectal Cancer via Smad2/3

**DOI:** 10.3390/cancers17030551

**Published:** 2025-02-06

**Authors:** Canfeng Fan, Qiang Wang, Saki Kanei, Kyoka Kawabata, Hinano Nishikubo, Rika Aoyama, Zhonglin Zhu, Daiki Imanishi, Takashi Sakuma, Koji Maruo, Gen Tsujio, Yurie Yamamoto, Tatsunari Fukuoka, Masakazu Yashiro

**Affiliations:** 1Molecular Oncology and Therapeutics, Osaka Metropolitan University Graduate School of Medicine, 1-4-3 Asahimachi, Abeno-ku, Osaka 545-8585, Japan; sy23126n@st.omu.ac.jp (C.F.); qiangw@vip.126.com (Q.W.); sn23107b@st.omu.ac.jp (S.K.); sn23131y@st.omu.ac.jp (K.K.); sn23089k@st.omu.ac.jp (H.N.); si22406x@st.omu.ac.jp (R.A.); w23032h@omu.ac.jp (Z.Z.); sy23003h@st.omu.ac.jp (D.I.); so22500y@st.omu.ac.jp (T.S.); r21764m@omu.ac.jp (K.M.); t-gen@leto.eonet.ne.jp (G.T.); f21269a@omu.ac.jp (Y.Y.); tfukuoka@omu.ac.jp (T.F.); 2Cancer Center for Translational Research, Osaka Metropolitan University Graduate School of Medicine, 1-4-3 Asahimachi, Abeno-ku, Osaka 545-8585, Japan; 3Department of Gastroenterological Surgery, Osaka Metropolitan University Graduate School of Medicine, 1-4-3 Asahimachi, Abeno-ku, Osaka 545-8585, Japan

**Keywords:** periostin, Smad2/3, cancer-associated fibroblasts, colorectal cancer, extracellular protein, stroma

## Abstract

Periostin has been shown to be abnormally expressed in cancer-associated fibroblasts (CAFs) across various cancer types and is regulated by the TGFβ/Smad signaling pathway. However, the patterns of periostin and Smad2/3 expression and their relationship in colorectal cancer have not yet been reported. In this study, we analyzed tissue samples from 351 colorectal cancer patients and found that periostin is predominantly expressed in CAFs rather than cancer cells. Moreover, its expression is strongly associated with Smad2/3 expression in CAFs.

## 1. Introduction

The tumor microenvironment characterized by an aberrant extracellular matrix (ECM) plays a crucial role in the progression of cancer [1]. Cancer-associated fibroblasts (CAFs) are instrumental in the remodeling of the extracellular matrix (ECM), the secretion of ECM proteins, and the facilitation of complex interactions between the tumor and its surrounding stroma [2]. Periostin, a matricellular protein encoded by the osteoblast-specific factor 2 gene (OSF-2), displays varied expression in normal human tissues, with marked expression in skin, breast, and ovarian tissues [3,4]. The expression of periostin is elevated in certain pathological circumstances, such as inflammatory conditions [5], and periostin is frequently overexpressed in the tumor stroma of some types of carcinomas, including prostate cancer, lung cancer, colorectal cancer, and pancreatic cancer [6,7,8,9,10,11,12]. Periostin has been reported to be synthesized by either cancer cells or surrounding tumor stromal cells, such as CAFs, within the tumor microenvironment [13].

The family of Smad proteins, comprising eight distinct types, is functionally categorized into three classes: R-Smad, Co-Smad, and I-Smad. Smad2 and Smad3, members of the R-Smad group, are primarily activated by tumor growth factor beta (TGFβ), and they function as transcriptional modulators that transduce and regulate gene expressions in response to TGFβ signaling [14]. The activation of a TGFβ/Smad pathway was reported to lead to an increase in the secretion of periostin in hepatocellular carcinoma CAFs [15], and it was proposed that prolonged abnormal activation of TGFβ in tumor tissues may increase the expression of Smad2/3, resulting in enhanced periostin expression [16,17].

Angiogenesis, motility, invasion, and immune response are partially regulated by TGF-β [15]. TGF-β signaling pathway associated with angiogenesis in various phases of carcinoma, such as carcinogenesis, tumor growth, and distant metastasis via nuclear translocation of Smad2 [15]. Although periostin and Smad2/3 signaling have been independently linked to cancer progression, their expression patterns and correlations remain insufficiently understood in colorectal cancer (CRC). Therefore, this study aimed to clarify the clinicopathologic significance of periostin and Smad2/3 expression in CRC, with a particular focus on the tumor microenvironment.

## 2. Materials and Methods

### 2.1. Patients

This study retrospectively analyzed a total of 351 CRC patients who received colectomy at Osaka Metropolitan University Hospital. The CRC tissues were obtained from 351 patients, including 49 patients with rectal cancers. The 351 patients did not receive cancer-related treatment, such as chemotherapy and radiotherapy, before surgery. The pathological diagnosis and classifications were performed in accordance with the revised 8th edition of the UICC classification, published in 2020. This study was approved by the Osaka Metropolitan University Ethics Committee (approval number 0924, 2022-077). The written informed consent for research was obtained from all patients. The inclusion criteria were histological diagnosis of colorectal cancer, having a complete medical history and clinical examination data, and having undergone surgery. The exclusion criteria were receiving any cancer-related treatment before surgery.

### 2.2. Immunostaining of Periostin and Smad2/3

The immunohistochemical determination of periostin and Smad2/3 expression was performed using tumor specimens of 351 CRC patients. Briefly, each slide was deparaffinized and heated for 40 min at 98 °C by a heater in Target Retrieval Solution High pH (DAKO, Carpinteria, CA, USA). After blocking endogenous peroxidase activity, the samples were incubated with an anti-POSTN antibody (1:200; Sigma Life Science, St. Louis, MO, USA) and an anti-Smad2/3 antibody (1:500, Santa Cruz, Dallas, TX, USA) for 24 h at 4 °C. These sections were incubated with biotinylated secondary antibody, followed by treatment with a streptavidin-peroxidase reagent, and were counterstained with Mayer’s hematoxylin.

### 2.3. Immunohistochemical Determination

The expression levels of periostin within CAFs, as well as in cancer cells, were analyzed by evaluating the staining intensity and the percentage of positively stained areas. The expression levels of Smad2/3 within CAFs, as well as in cancer cells, were analyzed by evaluating the staining intensity and the percentage of positively stained areas. Periostin and Smad2/3 immunostaining levels were evaluated as follows: the intensity was given scores 0–3 (0 = negative, 1+ = weak, 2+ = moderate, 3+ = intense), and the positivity in cancer cells or CAFs was also graded on a scale from 0 to 3 (0 = 0%–1%, 1 = 2%–30%, 2 = 31%–60%, 3 = 61%–100%). The two scores were added to obtain the results of 0–6. The expression was considered positive when the scores were ≥5 for periostin and ≥4 for Smad2/3. The immunostaining levels were evaluated by two independent observers who were unaware of clinical data and outcomes. When a different evaluation between the two independent observers was found, the evaluation was rechecked and discussed.

### 2.4. Statistical Analysis

We used the χ2 test or Fisher’s exact to decide the significance of the difference between covariates. The survival durations were calculated using the Kaplan–Meier method and analyzed by the log-rank test to compare the cumulative survival durations in the patient groups. In addition, the Cox proportional hazards model was used to compute univariate hazard ratios for the study parameters. The R statistical software (version 4.3.2; R Foundation for Statistical Computing, Vienna, Austria) was used for the analyses. A *p*-value < 0.05 indicated a statistically significant difference.

## 3. Results

### 3.1. Immunostaining of Periostin and Smad2/3

We observed that periostin was localized predominantly at tumor stroma, including the ECM and the cytoplasm of CAFs (Figure 1) and less expressed on the cancer cells. Periostin expression was observed mainly in CAFs of 129 (36.8%) of all 351 CRC cases, and on cancer cells in only 2 (0.6%) of the 351 CRC cases. In the tumor stroma, Smad2/3 was expressed primarily in the cytoplasm or nucleus of CAFs rather than in the ECM (Figure 1). A higher proportion of cancer cells exhibited elevated expression of Smad2/3. The expression of Smad2/3 was observed mainly in CAFs in 144 cases (41.0%) and on cancer cells in 316 (90.0%) cases.

### 3.2. The Relationship Between Clinicopathological Features and Periostin and Smad2/3 Expression

The clinicopathological characteristics of all 351 cases of CRC based on their periostin and Smad2/3 expression are summarized in Table 1. The periostin-positive stroma group was significantly correlated with a high T-stage (*p* = 0.011), lymph node metastasis (*p* = 0.011), venous invasion (*p* = 0.006), a high relapse rate (*p* = 0.015), and a high advanced stage (*p* = 0.015). The Smad2/3-positive cancer cell group (n = 144) was significantly correlated with a high relapse rate (*p* = 0.032), a high advanced stage (*p* = 0.049), and periostin positivity (*p* < 0.001). Both the periostin-positive tumor stroma group and the Smad2/3-positive cancer cells group showed significant correlations with venous invasion (*p* = 0.023), a high relapse rate (*p* = 0.014), and a high advanced stage (*p* = 0.035).

### 3.3. The Correlation Between the Patients’ Survival and the Periostin and/or Smad2/3 Expression

Figure 2 shows that Kaplan–Meier survival curves indicated that the 5-year overall survival rate of the patients in the periostin expression-, Smad2/3 expression-, and both-expression-positive groups was significantly (*p* < 0.01) worse compared to that of the negative group. As in prior studies of clinicopathological correlations, in stage 3 compared to stage 2, the patient group that was positive for periostin expression, Smad2/3 expression, and their co-expression was significantly associated with poorer survival rates compared to the negative group (Figure 2).

### 3.4. Univariate and Multivariate Analyses of Overall Survival

Table 2 provides the univariate and multivariate analyses of the patients’ overall survival. The univariate analysis indicated that poor survival was significantly (*p* < 0.001) correlated with periostin-positive expression, Smad2/3-positive expression (*p* < 0.001), age ≥ 65 years (*p* = 0.021), poorer differentiation (*p* < 0.001), lymph node metastasis (*p* < 0.001), lymphatic invasion (*p* = 0.010), and venous invasion (*p* = 0.010). Among these clinicopathologic factors, the multivariate analysis revealed that periostin in CAFs-positive expression (*p* < 0.001), age ≥ 65 years (*p* = 0.006), poorer differentiation (*p* = 0.001), and lymph node metastasis (*p* = 0.019) were significantly correlated with poor survival.

In the univariate analysis of the patients with stage 2 CRC (n = 191), only periostin-positive expression (*p* = 0.036) was significantly correlated with poorer survival. In the stage 3 group (n = 160), the univariate analysis revealed that poor survival was significantly correlated with periostin-positive expression (*p* < 0.001), Smad2/3-positive expression (*p* = 0.003), age ≥ 65 years (*p* = 0.012), and poorer differentiation (*p* = 0.009). The multivariate analysis revealed that the factors that were significantly correlated with poor survival were periostin in CAFs-positive expression (*p* = 0.004), age ≥ 65 years (*p* = 0.004), and poorer differentiation (*p* = 0.002).

## 4. Discussion

In this study, periostin expression was observed mainly in CAFs of 129 (36.8%) of all 351 CRC cases and on cancer cells in only 2 (0.6%) of the 351 CRC cases. A previous study reported that periostin expression was found in 27%–59% of CRC cases, while periostin is not expressed in normal colorectal tissues, and that epithelial cells show only weak or undetectable expression in stromal cells [18,19,20,21,22]. In our 351 CRC cases, periostin expression was mainly found in stromal cells. In addition, CRC with periostin-positive expression in tumor stromal cells was significantly correlated with lymph node metastasis, venous invasion, and a high relapse rate. These findings might suggest that periostin is produced mainly by CAFs and affects the malignant progression of CRC cells. It might be important to clarify the periostin production from CAF and the effect of periostin on the invasion activity of CRC cells, using culturing cells such as fibroblasts and cancer cell lines, in future studies.

Smad2/3 was reported to bind to the promoter region of periostin, regulate its expression, and enhance periostin levels [17,23]. We observed Smad2/3 expression in CAFs in 144 (41.0%) of 351 cases of CRC. Smad2/3-positive expression in CAFs was correlated with periostin expression in CAFs. In addition, the CRC patients with both periostin-positive and Smad2/3-positive CAFs were closely associated with venous invasion, tumor relapse, and a highly advanced stage. These findings might indicate that the periostin expression in CAFs is regulated via TGFβ/Smad signaling.

In this study, we observed a strong correlation between Smad2/3 and periostin expression in fibroblasts. However, in cancer cells, despite a high Smad2/3 positivity rate of 90%, periostin expression was notably low at only 0.6%. Previous reports have indicated that smad2/3, after translocating into the nucleus, interacts with various transcription factors to regulate specific target genes. In cancer cells, Smad2/3 preferentially regulates genes involved in cell proliferation. However, the specific transcription factors that Smad2/3 collaborates with to regulate periostin in CAFs remain largely unexplored and have yet to be clearly elucidated [24]. The primary function of CAFs is to shape the tumor microenvironment by secreting ECM proteins, such as periostin, and metabolic factors, thereby enhancing cancer cell survival and migratory capacity. In contrast, cancer cells focus primarily on their own proliferation and invasive capabilities, relatively independent of microenvironment regulation. Consequently, although Smad2/3 signaling is highly active in cancer cells, its target genes differ from those regulated by Smad2/3 in CAFs, such as periostin.

The Kaplan–Meier survival curves in the present population demonstrated that the CRC patients with the co-expression of periostin and smad2/3 had a significantly lower survival rate. This association with a lower 5-year overall survival rate was more pronounced in the stage 3 patients compared to those with stage 2 disease, and the multivariate analysis revealed that periostin expression serves as an independent prognostic factor to survival. These results indicate that periostin may serve as a useful independent prognostic indicator for patients with CRC, especially among those at an advanced stage. TGFβ exerts a dual regulatory effect of suppressing cancer cell growth in the early stages and promoting cancer cell invasion and metastasis in the later stages [25,26].

Our data indicate that the expression of periostin in the CAFs was not correlated with potentially aggressive cancer characteristics that are present at stage 2 and showed a relationship only at stage 3. In contrast, we observed that the expression of periostin in CAFs was highly correlated with Smad2/3 expression in CAFs from both stage 2 and stage 3 patients. These results suggested that periostin from CAFs, regulated by the TGFβ/Smad signaling pathway, might play an important role in the malignant progression of CRC cells, especially at advanced stages.

It was reported that periostin can enhance the stemness of ovarian cancer cells, which may be a crucial factor in the high expression of periostin that is associated with aggressive cancer characteristics [27]. At the advanced stages of tumor progression, periostin promotes the metastatic development of tumors by binding to αvβ3 integrins, thereby activating the Akt/PKB cell survival signaling pathway [28]. It has also been reported that integrin expression is elevated in late-stage colorectal cancer, which may explain the findings in this study [29,30]. It has been suggested that periostin secreted by stromal cells interacts with integrins (such as αvβ3 or αvβ5) on the surface of cancer cells to stimulate the FAK/ERK pathway, thereby increasing the secretion of TGFβ. This TGFβ then binds to receptors on stromal cells, activating the TGFβ/Smad pathway, which further enhances periostin secretion, forming a feedback loop of crosstalk. Periostin–integrin binding has been shown to play a critical role in promoting cancer cell migration and invasion, with the FAK/ERK pathway serving as a central mediator of these effects [10]. Additionally, the TGFβ/Smad pathway in stromal cells not only regulates periostin secretion but also contributes to ECM remodeling and stromal activation. Despite this understanding, the precise molecular events within this feedback loop remain unclear. The disruption of periostin’s interaction with cellular surface receptors on cancer cells may represent a viable therapeutic strategy and could potentially disrupt the cross-talk between cancer cells and CAFs. Our findings suggest that periostin might be a promising molecular target for CRC.

In conclusion, periostin from cancer-associated fibroblasts might be associated with the malignant progression of CRC via Smad2/3 signaling.

## 5. Conclusions

This work aims to clarify the expression patterns of periostin and Smad2/3 signaling in colorectal cancer and their relationship with clinicopathological features. Our results demonstrate that periostin is strongly expressed in cancer-associated fibroblasts (CAFs) and is closely associated with Smad2/3 expression, as well as malignant clinicopathological characteristics in colorectal cancer. Therefore, targeting periostin expression in CAFs by modulating the Smad2/3 pathway may represent a promising therapeutic approach.

## Figures and Tables

**Figure 1 cancers-17-00551-f001:**
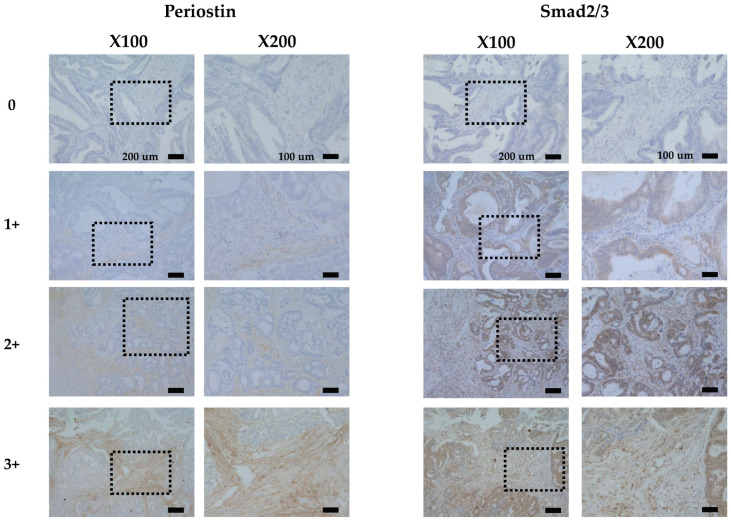
Representative cases showing the expressions of periostin and Smad2/3 expression in colorectal cancer, stained with hematoxylin and eosin (H&E). Periostin was expressed mainly in the cytoplasm of stromal cells. Smad2/3 was expressed mainly in the cytoplasm of cancer cells. The immunoreactivity of periostin and Smad2/3 was evaluated according to the intensity of staining categorized into four levels: 0 = negative, 1 = weak moderate, 2 = moderate, and 3 = intense.

**Figure 2 cancers-17-00551-f002:**
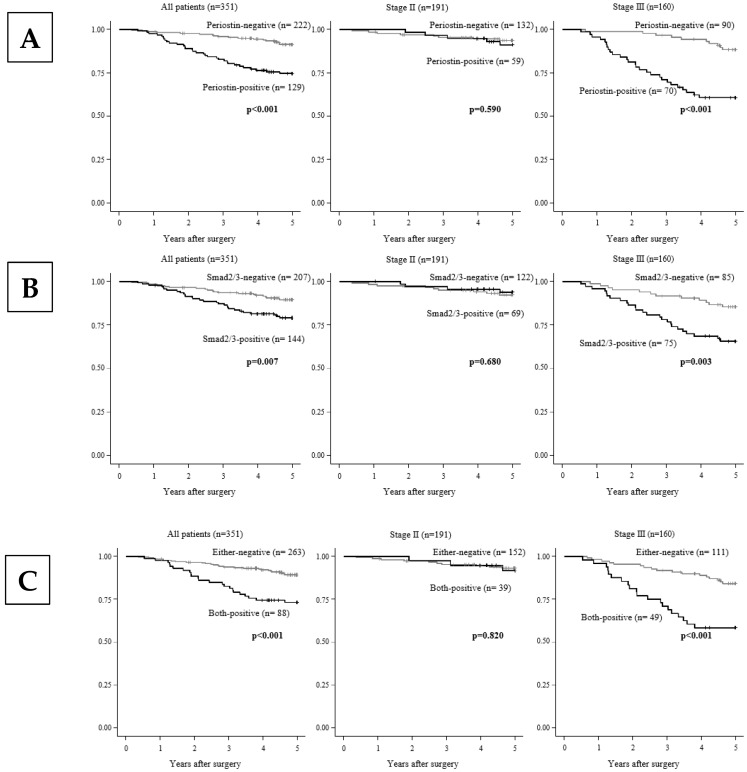
Kaplan–Meier survival curves. (**A**) The overall survival (OS) of the patients with periostin-positive tumor stroma was significantly poorer than that of the patients with periostin-negative tumors. Among the patients with stage 3 disease, the OS of those with periostin-positive tumor stroma was significantly poorer than that of the patients with periostin-negative tumors (*p* < 0.001), and positive periostin expression was significantly associated with poorer survival rates compared to the negative-expression group. (**B**) The OS of the patients with Smad2/3-positive tumors was significantly poorer than that of the patients with Smad2/3-negative tumors (*p* < 0.001). Among the stage 3 patients (*p* < 0.001) compared to stage 2 (*p* = 0.330), the patients who were positive for Smad2/3 expression showed significantly poorer OS compared to the Smad2/3-negative patients. (**C**) The OS of the patients with both-expression-positive tumors was significantly poorer than that of the patients with either-expression-negative tumors (*p* < 0.001). In stage 3 (*p* < 0.001) compared to stage 2 (*p* = 0.015), the positive group for co-expression was significantly associated with poorer OS compared to the negative group. Both-expression-positive: periostin+/Smad2/3+. Either-expression-negative: periostin−/Smad2/3−, periostin+/Smad2/3−, and periostin−/Smad2/3.

**Table 1 cancers-17-00551-t001:** Correlation between clinicopathological factors and the expression of Periostin, Smad2/3, and their co-expression in 351 colon tumor stroma.

All Cases										
Clinicopathologic Features		Periostin		*p*-Value	Smad2/3		*p*-Value	Co-Expression		*p*-Value
		Positive (n = 129)	Negative (n = 222)		Positive (n = 144)	Negative (n = 207)		Both-Positive (n = 88)	Either-Negative (n = 263)	
Age	<65	52 (38%)	85 (62%)	0.734	53 (39%)	84 (61%)	0.506	31 (23%)	106 (77%)	0.449
	≥65	77 (36%)	137 (64%)		91 (43%)	123 (57%)		57 (27%)	157 (73%)	
Gender	Female	56 (33%)	116 (67%)	0.122	73 (42%)	99 (58%)	0.664	38 (22%)	134 (78%)	0.22
	Male	73 (41%)	106 (59%)		71 (40%)	108 (60%)		50 (28%)	129 (72%)	
Histological type	Well	114 (35%)	209 (65%)	0.066	129 (40%)	194 (60%)	0.167	77 (24%)	246 (76%)	0.108
	Poor	15 (54%)	13 (46%)		15 (54%)	13 (46%)		11 (39%)	17 (61%)	
T stage	T1	72 (32%)	155 (68%)	0.011	94 (41%)	133 (59%)	0.91	52 (23%)	175 (77%)	0.246
	T2, T3, and T4	57 (46%)	67 (54%)		50 (40%)	74 (60%)		36 (29%)	88 (71%)	
Lymph node metastasis	Negative	59 (31%)	133 (69%)	0.011	70 (36%)	122 (64%)	0.064	39 (20%)	153 (80%)	0.026
	Positive	70 (44%)	89 (56%)		74 (47%)	85 (53%)		49 (31%)	110 (69%)	
Lymphatic invasion	Negative	28 (29%)	68 (71%)	0.082	36 (38%)	60 (62%)	0.466	18 (19%)	78 (81%)	0.099
	Positive	101 (40%)	154 (60%)		108 (42%)	147 (58%)		70 (27%)	185 (73%)	
Venous invasion	Negative	92 (33%)	186 (67%)	0.006	108 (39%)	170 (61%)	0.111	62 (22%)	216 (78%)	0.023
	Positive	37 (51%)	36 (49%)		36 (49%)	37 (51%)		26 (36%)	47 (64%)	
Location	Left	81 (34%)	156 (66%)	0.158	99 (42%)	138 (58%)	0.729	56 (24%)	181 (76%)	0.43
	Right	48 (42%)	66 (58%)		45 (39%)	69 (61%)		32 (28%)	82 (72%)	
Relapse	Negative	88 (33%)	179 (67%)	0.01	101 (38%)	166 (62%)	0.032	58 (22%)	209 (78%)	0.014
	Positive	41 (49%)	43 (51%)		43 (51%)	41 (49%)		30 (36%)	54 (64%)	
Stage	2	59 (31%)	132 (69%)	0.015	69 (36%)	122 (64%)	0.049	39 (20%)	152 (80%)	0.035
	3	70 (44%)	90 (56%)		75 (47%)	85 (53%)		49 (31%)	111 (69%)	
Periostin	Negative				56 (25%)	166 (75%)	<0.001			
	Positive				88 (68%)	41 (32%)				
Stage 2										
Clinicopathologic Features		Periostin		*p*-value	Smad2/3		*p*-value	Co-Expression		*p*-value
		Positive (n = 59)	Negative (n = 132)		Positive (n = 69)	Negative (n = 122)		Both-Positive (n = 39)	Either-Negative (n = 152)	
Age	<65	23 (32%)	50 (68%)	1	24 (33%)	49 (67%)	0.536	12 (16%)	61 (84%)	0.356
	≥65	36 (31%)	82 (69%)		45 (38%)	73 (62%)		27 (23%)	91 (77%)	
Gender	Female	23 (26%)	65 (74%)	0.211	35 (40%)	53 (60%)	0.366	15 (17%)	73 (83%)	0.368
	Male	36 (35%)	67 (65%)		34 (33%)	69 (67%)		24 (23%)	79 (77%)	
Histological type	Well	56 (30%)	128 (70%)	0.679	67 (36%)	117 (64%)	1	38 (21%)	146 (79%)	1
	Poor	3 (43%)	4 (57%)		2 (29%)	5 (71%)		1 (14%)	6 (86%)	
T stage	T1	37 (28%)	94 (72%)	0.244	50 (38%)	81 (62%)	0.421	26 (20%)	105 (80%)	0.847
	T2, T3, and T4	22 (37%)	38 (63%)		19 (32%)	41 (68%)		13 (22%)	47 (78%)	
Lymph node metastasis	Negative	59 (31%)	131 (69%)	1	69 (36%)	121 (64%)	1	39 (21%)	151 (79%)	1
	Positive	0 (0%)	1 (100%)		0 (0%)	1 (100%)		0 (0%)	1 (100%)	
Lymphatic invasion	Negative	22 (28%)	56 (72%)	0.528	27 (35%)	51 (65%)	0.761	13 (17%)	65 (83%)	0.362
	Positive	37 (33%)	76 (67%)		42 (37%)	71 (63%)		26 (23%)	87 (77%)	
Venous invasion	Negative	48 (30%)	113 (70%)	0.52	59 (37%)	102 (63%)	0.837	33 (20%)	128 (80%)	1
	Positive	11 (37%)	19 (63%)		10 (33%)	20 (67%)		6 (20%)	24 (80%)	
Location	Left	33 (27%)	91 (73%)	0.101	46 (37%)	78 (63%)	0.754	24 (19%)	100 (81%)	0.707
	Right	26 (39%)	41 (61%)		23 (34%)	44 (66%)		15 (22%)	52 (78%)	
Relapse	Negative	52 (31%)	115 (69%)	1	60 (36%)	107 (64%)	1	33 (20%)	134 (80%)	0.589
	Positive	7 (29%)	17 (71%)		9 (38%)	15 (62%)		6 (25%)	18 (75%)	
Periostin	Negative				30 (23%)	102 (77%)	<0.001			
Stage 3										
Clinicopathologic Features		Periostin		*p*-value	Smad2/3		*p*-value	Co-Expression		*p*-value
		Positive (n = 70)	Negative (n = 90)		Positive (n = 75)	Negative (n = 85)		Both-Positive (n = 49)	Either-Negative (n = 111)	
Age	<65	29 (45%)	35 (55%)	0.748	29 (45%)	35 (55%)	0.872	19 (30%)	45 (70%)	0.863
	≥65	41 (43%)	55 (57%)		46 (48%)	50 (52%)		30 (31%)	66 (69%)	
Gender	Female	33 (39%)	51 (61%)	0.265	38 (45%)	46 (55%)	0.751	23 (27%)	61 (73%)	0.393
	Male	37 (49%)	39 (51%)		37 (49%)	39 (51%)		26 (34%)	50 (66%)	
Histological type	Well	58 (42%)	81 (58%)	0.239	62 (45%)	77 (55%)	0.163	39 (28%)	100 (72%)	0.08
	Poor	12 (57%)	9 (43%)		13 (62%)	8 (38%)		10 (48%)	11 (52%)	
T stage	T1	35 (36%)	61 (64%)	0.034	44 (46%)	52 (54%)	0.75	26 (27%)	70 (73%)	0.294
	T2, T3, and T4	35 (55%)	29 (45%)		31 (48%)	33 (52%)		23 (36%)	41 (64%)	
Lymph node metastasis	Negative	0 (0%)	2 (100%)	0.505	1 (50%)	1 (50%)	1	0 (0%)	2 (100%)	1
	Positive	70 (44%)	88 (56%)		74 (47%)	84 (53%)		49 (31%)	109 (69%)	
Lymphatic invasion	Negative	6 (33%)	12 (67%)	0.452	9 (50%)	9 (50%)	0.807	5 (28%)	13 (72%)	1
	Positive	64 (45%)	78 (55%)		66 (46%)	76 (54%)		44 (31%)	98 (69%)	
Venous invasion	Negative	44 (38%)	73 (62%)	0.012	49 (42%)	68 (58%)	0.049	29 (25%)	88 (75%)	0.012
	Positive	26 (60%)	17 (40%)		26 (60%)	17 (40%)		20 (47%)	23 (53%)	
Location	Left	48 (42%)	65 (58%)	0.727	53 (47%)	60 (53%)	1	32 (28%)	81 (72%)	0.35
	Right	22 (47%)	25 (53%)		22 (47%)	25 (53%)		17 (36%)	30 (64%)	
Relapse	Negative	36 (36%)	64 (64%)	0.014	41 (41%)	59 (59%)	0.072	25 (25%)	75 (75%)	0.053
	Positive	34 (57%)	26 (43%)		34 (57%)	26 (43%)		24 (40%)	36 (60%)	
Periostin	Negative				26 (29%)	64 (71%)	<0.001			
	Positive				49 (70%)	21 (30%)				

Histological type: well, well-differentiated tumors; poor, poorly differentiated tumors.

**Table 2 cancers-17-00551-t002:** Univariate and multivariate Cox multiple regression analysis with respect to overall survival after surgery.

All Cases						
	Univariate			Multivariate		
Parameter	Hazard Ratio	95%CI	*p*-Value	Hazard Ratio	95%CI	*p*-Value
Periostin	2.949	1.851–4.697	<0.001	2.518	1.475–4.299	<0.001
Smad2/3	2.214	1.389–3.53	<0.001	1.291	0.757–2.201	0.348
Age	1.805	1.091–2.987	0.021	2.045	1.222–3.42	0.006
Gender	1.031	0.653–1.63	0.895			
Histological type	2.91	1.597–5.302	<0.001	2.76	1.484–5.133	0.001
T stage	1.233	0.773–1.968	0.38			
Lymph node metastasis	2.422	1.507–3.893	<0.001	1.832	1.104–3.042	0.019
Lymphatic invasion	2.317	1.22–4.398	0.01	1.639	0.834–3.219	0.152
Venous invasion	1.91	1.167–3.127	0.01	1.215	0.721–2.049	0.465
Location	1.196	0.738–1.938	0.467			
Stage 2						
Parameter	Hazard Ratio	95%CI	*p*-Value			
Periostin	2.336	1.057–5.163	0.036			
Smad2/3	1.507	0.661–3.438	0.33			
Age	1.371	0.595–3.161	0.459			
Gender	0.625	0.282–1.385	0.247			
Histological type	1.928	0.45–8.262	0.377			
T stage	0.862	0.359–2.073	0.74			
Lymph node metastasis	0	0-Inf	0.997			
Lymphatic invasion	1.161	0.51–2.64	0.722			
Venous invasion	2.029	0.807–5.097	0.132			
Location	1.005	0.432–2.338	0.99			
Stage 3						
Parameter	Hazard Ratio	95%CI	*p*-Value	Hazard Ratio	95%CI	*p*-Value
Periostin	2.964	1.654–5.313	<0.001	2.681	1.381–5.207	0.004
Smad2/3	2.382	1.33–4.265	0.003	1.454	0.749–2.823	0.268
Age	2.259	1.197–4.264	0.012	2.55	1.341–4.852	0.004
Gender	1.517	0.864–2.665	0.147			
Histological type	2.456	1.254–4.811	0.009	2.868	1.449–5.676	0.002
T stage	1.307	0.744–2.297	0.351			
Lymph node metastasis	0.329	0.08–1.364	0.126			
Lymphatic invasion	3.538	0.859–14.569	0.08			
Venous invasion	1.524	0.845–2.747	0.161			
Location	1.503	0.834–2.71	0.175			

## Data Availability

The datasets generated during and/or analyzed during the current study are available from the corresponding author upon reasonable request.

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
