# Peer review of "Periostin from Tumor Stromal Cells Might Be Associated with Malignant Progression of Colorectal Cancer via Smad2/3"

_cancers, 2025, doi:10.3390/cancers17030551_

Round 1

Reviewer 1 Report

Comments and Suggestions for Authors

The study aims to explore the association between periostin expression from tumor stromal cells and the malignant progression of colorectal cancer. A total of 351 tissue samples were analyzed for periostin and/or Smad2/3 expression, and overall survival was compared between periostin-positive and periostin-negative tumors.

However, the evidence provided is insufficient to confirm this association. Periostin was expressed in only 0.6% of cancer cells, suggesting limited direct involvement. If malignant progression is driven by periostin-expressing cancer-associated fibroblasts (CAFs), this requires experimental validation, which is not provided.

Additionally, periostin expression was not analyzed across different cancer stages or compared with normal tissues, leaving gaps in understanding its role. Without further data and functional studies, the conclusions remain unsupported.

Periostin and Smad2/3 expression levels were shown in colorectal cancer tissues, but the figures lack clarity for meaningful comparisons. The scale bars should be legible to indicate the magnification clearly.

In summary, while the study raises an important question, additional experimental evidence and comparative analyses are needed to clarify periostin’s role in colorectal cancer progression.

Author Response

We have responded to Referee #1 comments, as follows.

Referee #1

Thank you very much for the careful reviews of the Referee #1. We correct several points according to the descriptions by the reviewer, as described below. We indicate the changes point by point and highlighted them in the revised paper.

  1. The evidence provided is insufficient to confirm this association. Periostin was expressed in only 0.6% of cancer cells, suggesting limited direct involvement. If malignant progression is driven by periostin-expressing cancer-associated fibroblasts (CAFs), this requires experimental validation, which is not provided. Additionally, periostin expression was not analyzed across different cancer stages or compared with normal tissues, leaving gaps in understanding its role. Without further data and functional studies, the conclusions remain unsupported.

In this study, periostin expression was observed mainly in CAFs of 129 (36.8%) of all 351 CRC cases, and on cancer cells in only two (0.6%) of the 351 CRC cases. Previous study reported that periostin expression was found at 27%–59% of CRC cases, while peri-ostin is not expressed in normal colorectal tissues, and that epithelial cells and shows on-ly weak or undetectable expression in stromal cells [18-22]. In our 351 CRC case, periostin expression was mainly found at stromal cells. In addition, CRC with periostin-positive expression at tumor stromal cells was significantly correlated with lymph node metastasis, venous invasion, and a high relapse rate. These findings might suggest that periostin is produced mainly by CAFs and affect the malignant progression of CRC cells. It might be important to clarify the periostin production from CAF and the effect of periostin on the invasion activity of CRC cells, using culturing cells such as fibroblasts and cancer cell lines, in future study. (Line 171-182)

  1. Periostin and Smad2/3 expression levels were shown in colorectal cancer tissues, but the figures lack clarity for meaningful comparisons. The scale bars should be legible to indicate the magnification clearly.

We added scale bars and labeled the actual size of the scale bars directly (Figure 1).

  1. In summary, while the study raises an important question, additional experimental evidence and comparative analyses are needed to clarify periostin’s role in colorectal cancer progression.

I agree with the reviewer comment. It is important to clarify the periostin’s role in colorectal cancer progression. As the near future study, we should examine the periostin’s role in colorectal cancer, using culturing cells such as fibroblasts and colorectal cancer cell lines.

Reviewer 2 Report

Comments and Suggestions for Authors

Title  should be modified eg. - Periostin from Tumor Stromal Cells Might Be Associated with Poor Outcome of Colorectal Cancer via Smad2/3

Abstract. Please provide here detailed data of Cox analysis.

Introduction. Please provide more information concerning an impact of TGF-beta on angiogenesis and modulation of inflammatory response.

The study  purpose should be enhanced.

Line 71 what number UICC TNM did you use. All cases were re-diagnosed. I did not find information about inclusion and exclusion criteria. What about chemotherapy in III stage?

Did you perform immunohistochemistry manually? Did you use any positive/negative control? Why did you apply the mentioned antibodies concentration.

Results.  Did you find the periostin expression in other ECM cells excluding CAFs? How did you fix that problem?

Table 1. I am confused with ‘T’ attribute lines – check it. T should be written in capitals. What means histological type -well/ poor?

Location – did you find any rectal cancer?  If yes it should be mentioned ; what about radiotherapy?

Table 2. What means T invasion?

Author Response

Referee #2

Thank you very much for the careful reviews of Referee #2. We correct several points according to the descriptions by the reviewer, as described below. We indicated the changes point by point and highlighted them in the revised paper.

  1. Title should be modified eg. - Periostin from Tumor Stromal Cells Might Be Associated with Poor Outcome of Colorectal Cancer via Smad2/3?

We revised the title to make it more concise and aligned with the content of the study, as follows; "Periostin from Tumor Stromal Cells Is Associated with Poor Outcome of Colorectal Cancer via Smad2/3". (Line 2-3)

  1. Abstract. Please provide here detailed data of Cox analysis.

We added detailed data of Cox analysis in Abstract section, as follows. In the stage 3 group (n=160) multivariate analysis revealed that periostin was an independent prognostic factor, while univariate analysis showed that both periostin and Smad2/3 were significantly correlated with poor survival. (Line 37-39)

  1. Introduction. Please provide more information concerning an impact of TGF-beta on angiogenesis and modulation of inflammatory response.

We added the information concerning an impact of TGF-beta in Introduction section, as follows. Angiogenesis, motility, invasion, and immune response are partially regulated by TGF-β [15]. TGF-β signaling pathway associated with angiogenesis in various phase of carcinoma such as carcinogenesis, tumor growth, and distant metastasis via nuclear translocation of Smad2 [15]. (Line 66-69)

  1. The study purpose should be enhanced.

We added the purpose in Introduction section, as follows. Although periostin and Smad2/3 signaling have been independently linked to cancer progression, their expression patterns and correlations remain to be insufficiently understood in colorectal cancer (CRC). Then, this study aimed to clarify the clinicopathologic significance of periostin and Smad2/3 expression in CRC, with a particular focus on the tumor microenvironment. We added the aim in Abstract section, as follows. In this study, we aimed to clarify the clinicopathologic significance of periostin and Smad2/3 expression in CRC, with a particular focus on the tumor microenvironment. (Line 69-73)

  1. Line 71 what number UICC TNM did you use. All cases were re-diagnosed. I did not find information about inclusion and exclusion criteria. What about chemotherapy in III stage?

The pathological diagnosis and classifications were performed in accordance with the revised 8th edition of the UICC classification, published in 2020. This study was approved by the Osaka Metropolitan University Ethics Committee (approval number 0924, 2022-077). The written informed consent for research was obtained from all patients. The inclusion criteria were histological diagnosis of colorectal cancer, have complete medical history and clinical examination data, and having undergone surgery. The exclusion criteria were had been receiving any cancer-related treatment before enrollment. (Line 80-83)

  1. Did you perform immunohistochemistry manually? Did you use any positive/negative control? Why did you apply the mentioned antibodies concentration.

Yes, we performed immunohistochemistry manually instead of using an automated immunohistochemistry system. Since the positive and negative regions were clearly distinguishable within the patient tissue samples, we did not include additional positive or negative controls. We tested three antibody concentrations as1:100, 1:200, and 1:500, and selected the concentration that provided clear staining results while minimizing high-intensity background signals.

  1. Results. Did you find the periostin expression in other ECM cells excluding CAFs? How did you fix that problem?

In our experiments, no periostin staining was observed in immune cells or vascular endothelial cells, both of which are readily identifiable. Other stromal cell types in the tumor microenvironment are less abundant, which may contribute to the lack of detectable staining in those cells. These findings suggested that the periostin expression observed was primarily localized to CAFs, consistent with their role as a major source of periostin in the stroma. In future study It might be important to clarify the periostin production from CAF and the effect of periostin on the invasion activity of CRC cells, using culturing cells such as fibroblasts and cancer cell lines. (Line 177-181)

  1. Table 1. I am confused with ‘T’ attribute lines – check it. T should be written in capitals. What means histological type -well/ poor?

We corrected 'T' in capitals. "well" refers to well-differentiated tumors, including cases such as tubular adenocarcinoma with a high degree of glandular formation. "Poor" refers to poorly differentiated tumors, which typically show minimal or no glandular structures, such as poorly-differentiated adenocarcinoma. These terms describe in table. (Table I) (Line 141)

  1. Location – did you find any rectal cancer? If yes it should be mentioned ; what about radiotherapy?

The CRC tissues were obtained from 351 patients including 49 patients with rectal cancers. The 351 patients have not received cancer-related treatment such as chemotherapy and radiotherapy before surgery. (Line 78-79)

10.Table 2. What means T invasion?

We mean "T invasion" as the "T stage", which represents the tumor invasion depth according to the TNM classification. We corrected "T invasion" to "T stage" in Table II.

Reviewer 3 Report

Comments and Suggestions for Authors

General Comments. The manuscript addresses the clinicopathological significance of periostin and Smad2/3 expression and their association with CRC progression. The authors employ a large patient cohort (n = 351), providing robust platforms and data supporting their conclusions. However, some aspects need further clarification and discussion.

Major comments:

  1. The authors observed a positive correlation between periostin and Smad2/3 expression predominantly in CAFs rather than cancer cells. The paper notes that while Smad2/3 expression was high in both CAFs (41.0%) and cancer cells (90.0%), periostin expression was almost exclusive to CAFs (36.8%), with negligible expression in cancer cells (0.6%). This suggests that the interaction between periostin and Smad2/3 primarily occurs in CAFs. However, the paper does not clearly discuss the importance of Smad2/3 in cancer cells versus CAFs in regulating periostin expression. Could the authors provide additional details or hypothesize on the mechanisms of CAF-cancer cell crosstalk mediated by periostin and Smad2/3 signaling?

  1. A quick analysis of POSTN (periostin), SMAD2, and SMAD3 expression (RNA-seq) from TCGA data of CRC patients shows the following average TPM values: (i) POSTN: 91.3, (ii) SMAD2: 36.2, and (iii) SMAD3: 24.7. These results suggest that periostin is expressed in CRC cells at higher levels compared to SMAD2/3. However, this contrasts with the authors' findings, where only 0.6% of CRC cells expressed periostin based on immunohistochemistry. Could the authors address this discrepancy and discuss potential reasons, such as differences in experimental methods, tumor microenvironment contributions, or CAF contamination in RNA-seq datasets?

  1. Within the 351 samples analyzed, were any adjacent "normal" tissues included to evaluate periostin and Smad2/3 expression in normal fibroblasts? This comparison could help confirm whether the observed overexpression of periostin and Smad2/3 is specific to CAFs or also present in normal fibroblasts. Additionally, could the authors discuss how periostin and Smad2/3 overexpression is specifically linked to CAFs and whether this association distinguishes them from normal fibroblasts?

  1. In the “2.3. Immunohistochemical determination” section, the authors described that periostin and Smad2/3 expression were scored based on intensity (0–3: none to intense) and positivity (0–3: 0–100%) in CRC cells or CAFs, with combined scores deemed positive at ≥5 for periostin and ≥4 for Smad2/3. Could the authors clarify the rationale behind selecting these specific thresholds and discuss how they might impact the statistical outcomes of the study?

Minor comments:

  1. The phrase "might be associated" in the title lacks scientific precision and undermines the strength of the study's findings. The title should reflect the evidence more conclusively while aligning with the study's core results.

  2. In the result section “3.2. The relationship between clinicopathological features and periostin/Smad23 expression”,  there is a typographical error where “Smad23” is written instead of “Smad2/3.” Additionally, in the same paragraph, the authors describe the “periostin-positive tumor group (line 115)” and the “Smad2/3-positive tumor group (line 115).” To avoid potential misinterpretation, they should clarify whether this refers to the tumor stroma, CAFs, or cancer cells, rather than using the term “tumor group” alone.

Author Response

Referee #3

Thank you very much for the careful reviews of Referee #3. We correct several points according to the descriptions by the reviewer, as described below. We indicated the changes point by point and highlighted them in the revised paper.

1.The authors observed a positive correlation between periostin and Smad2/3 expression predominantly in CAFs rather than cancer cells. The paper notes that while Smad2/3 expression was high in both CAFs (41.0%) and cancer cells (90.0%), periostin expression was almost exclusive to CAFs (36.8%), with negligible expression in cancer cells (0.6%). This suggests that the interaction between periostin and Smad2/3 primarily occurs in CAFs. However, the paper does not clearly discuss the importance of Smad2/3 in cancer cells versus CAFs in regulating periostin expression. Could the authors provide additional details or hypothesize on the mechanisms of CAF-cancer cell crosstalk mediated by periostin and Smad2/3 signaling?

We added the discussion about the reasons for the different expression patterns of SMAD2/3 and periostin in cancer cells and CAFs, as well as the crosstalk between cancer cells and CAFs, as follows.

Smad2/3 might interact with various transcription factors to regulate specific target genes after translocating into the nucleus. In cancer cells, Smad2/3 preferentially regulates genes involved in cell proliferation. Sincethe specific transcription factors that Smad2/3 collaborates with to regulate periostin in CAFs remain largely unexplored and have yet to be clearly elucidated, so far. The primary function of CAFs is to shape the tumor microenvironment by secreting ECM proteins, such as periostin, and metabolic factors, thereby enhancing cancer cell survival and migratory capacity. In contrast, cancer cells focus primarily on their own proliferation and invasive capabilities, relatively independent of microenvironment regulation. Consequently, although Smad2/3 signaling is highly active in cancer cells, its target genes differ from those regulated by Smad2/3 in CAFs, such as periostin. (Line 193-203)

It has been suggested that periostin secreted by stromal cells interacts with integrins (such as αvβ3 or αvβ5) on the surface of cancer cells to stimulate the FAK/ERK pathway, thereby increasing the secretion of TGFβ. This TGFβ then binds to receptors on stromal cells, acti-vating the TGFβ/Smad pathway, which further enhances periostin secretion, forming a feedback loop of crosstalk. Periostin-integrin binding has been shown to play a critical role in promoting cancer cell migration and invasion, with the FAK/ERK pathway serving as a central mediator of these effects. [10]. Additionally, the TGFβ/Smad pathway in stromal cells not only regulates periostin secretion but also contributes to ECM remodeling and stromal activation. Despite this understanding, the precise molecular events within this feedback loop remain unclear. (Line 226-235)

  1. A quick analysis of POSTN (periostin), SMAD2, and SMAD3 expression (RNA-seq) from TCGA data of CRC patients shows the following average TPM values: (i) POSTN: 91.3, (ii) SMAD2: 36.2, and (iii) SMAD3: 24.7. These results suggest that periostin is expressed in CRC cells at higher levels compared to SMAD2/3. However, this contrasts with the authors' findings, where only 0.6% of CRC cells expressed periostin based on immunohistochemistry. Could the authors address this discrepancy and discuss potential reasons, such as differences in experimental methods, tumor microenvironment contributions, or CAF contamination in RNA-seq datasets?

We recognize the discrepancy between the RNA-seq data from TCGA and our immunohistochemistry (IHC) findings. Then, we referred to the publicly available Single Cell Portal's Human Colon Cancer Atlas (c295) and conducted an additional analysis of POSTN expression. The single-cell RNA sequencing data clearly show that POSTN expression is almost exclusively confined to stromal cells, such as cancer-associated fibroblasts (CAFs), while epithelial (cancer) cells show minimal or no expression of POSTN as described below. These observations align with our IHC data, where only 0.6% of colorectal cancer cells expressed periostin. The higher TPM values for POSTN observed in TCGA bulk RNA-seq data are likely due to stromal cell contributions, as bulk sequencing captures RNA from all cell types within the tumor
microenvironment and does not differentiate between epithelial and stromal sources.

  1. Within the 351 samples analyzed, were any adjacent "normal" tissues included to evaluate periostin and Smad2/3 expression in normal fibroblasts? This comparison could help confirm whether the observed overexpression of periostin and Smad2/3 is specific to CAFs or also present in normal fibroblasts. Additionally, could the authors discuss how periostin and Smad2/3 overexpression is specifically linked to CAFs and whether this association distinguishes them from normal fibroblasts?

While we did not perform staining on adjacent normal tissues in our study, previous research has demonstrated that periostin expression is generally low or undetectable in normal epithelial cells and stromal fibroblasts of non-tumorous tissues. To address this point, we have revised the Discussion section as follows: In this study, periostin expression was observed mainly in CAFs of 129 (36.8%) of all 351 CRC cases, and on cancer cells in only two (0.6%) of the 351 CRC cases. Previous study reported that periostin expression was found at 27%–59% of CRC cases, while periostin is not expressed in normal colorectal tissues, and that epithelial cells and shows only weak or undetectable expression in stromal cells [18-22]. In our 351 CRC case, periostin expres-sion was mainly found at stromal cells. In addition, CRC with periostin-positive expres-sion at tumor stromal cells was significantly correlated with lymph node metastasis, ve-nous invasion, and a high relapse rate. These findings might suggest that periostin is produced mainly by CAFs and affect the malignant progression of CRC cells. It might be important to clarify the periostin production from CAF and the effect of periostin on the invasion activity of CRC cells, using culturing cells such as fibroblasts and cancer cell lines, in future study. (Line 171-182)

  1. In the “2.3. Immunohistochemical determination” section, the authors described that periostin and Smad2/3 expression were scored based on intensity (0–3: none to intense) and positivity (0–3: 0–100%) in CRC cells or CAFs, with combined scores deemed positive at ≥5 for periostin and ≥4 for Smad2/3. Could the authors clarify the rationale behind selecting these specific thresholds and discuss how they might impact the statistical outcomes of the study?

The thresholds for periostin (≥5) and Smad2/3 (≥4) were determined based on their ability to effectively distinguish positive and negative expression groups in our cohort. Preliminary analyses showed that these thresholds provided optimal separation of groups with distinct clinicopathological characteristics and outcomes, ensuring robust classification and meaningful comparisons. We acknowledge that different thresholds may have some influence on the results; however, slight variations in these values would not significantly alter the overall trends or conclusions. The strong correlations observed between periostin and Smad2/3 expression and clinical parameters indicate that the chosen thresholds reliably capture biologically and clinically meaningful differences. Then, we confirmed the rationale behind selecting these specific thresholds. These strong correlations observed between periostin and Smad2/3 expression might support the significance of periostin and Smad2/3 as therapeutic molecular targets for CRC. (Line 235-238)

  1. The phrase "might be associated" in the title lacks scientific precision and undermines the strength of the study's findings. The title should reflect the evidence more conclusively while aligning with the study's core results.

We revised the title to make it more concise and aligned with the content of the study, as follows; "Periostin from Tumor Stromal Cells Is Associated with Poor Outcome of Colorectal Cancer via Smad2/3". (Line 2-3)

  1. In the result section “3.2. The relationship between clinicopathological features and periostin/Smad23 expression”, there is a typographical error where “Smad23” is written instead of “Smad2/3.” Additionally, in the same paragraph, the authors describe the “periostin-positive tumor group (line 115)” and the “Smad2/3-positive tumor group (line 115).” To avoid potential misinterpretation, they should clarify whether this refers to the tumor stroma, CAFs, or cancer cells, rather than using the term “tumor group” alone.

We corrected a typographical error of Smad23 to Smad2/3. We corrected periostin-positive tumor group to periostin-positive tumor stroma group. We corrected Smad2/3-positive tumor group to periostin-positive cancer cells group.

Round 2

Reviewer 1 Report

Comments and Suggestions for Authors

The author's responses have sufficiently clarified my concerns. I find the manuscript suitable for acceptance in its current form.

Reviewer 2 Report

Comments and Suggestions for Authors

The manuscript is much better in the present form. I accept the revised version.

Reviewer 3 Report

Comments and Suggestions for Authors

The authors have effectively addressed the comments, significantly improving the paper's clarity and overall quality.